# Bacteriophage Therapy: Developments and Directions

**DOI:** 10.3390/antibiotics9030135

**Published:** 2020-03-24

**Authors:** Mikeljon P. Nikolich, Andrey A. Filippov

**Affiliations:** Wound Infections Department, Bacterial Diseases Branch, Walter Reed Army Institute of Research, Silver Spring, MD 20910, USA; andrey.a.filippov.ctr@mail.mil

**Keywords:** bacteriophage therapy, antimicrobial, multidrug resistance, phage cocktail

## Abstract

In an era of proliferating multidrug resistant bacterial infections that are exhausting the capacity of existing chemical antibiotics and in which the development of new antibiotics is significantly rarer, Western medicine must seek additional therapeutic options that can be employed to treat these infections. Among the potential antibacterial solutions are bacteriophage therapeutics, which possess very different properties from broad spectrum antibiotics that are currently the standard of care, and which can be used in combination with them and often provide synergies. In this review we summarize the state of the development of bacteriophage therapeutics and discuss potential paths to the implementation of phage therapies in contemporary medicine, focused on fixed phage cocktail therapeutics since these are likely to be the first bacteriophage products licensed for broad use in Western countries.

## 1. Introduction

The use of the viruses of bacteria, bacteriophages (phages), as therapeutic agents to treat bacterial infections began 20 years before the first clinical use of an antibiotic drug, but the introduction of broad-spectrum antibiotics in the 1940s rapidly eclipsed and displaced the development of phage therapeutics in much of the world. Phage therapy continued to be developed with ongoing use in the Soviet Union and Eastern Europe that continues to the present in Poland, Russia, and Georgia [1,2,3]. However, until the rising global dissemination of multidrug-resistant (MDR) bacterial infections [4], earnest interest in clinical application was not evident in Western Europe and the Americas, nor in much of the rest of the world. Multidrug resistance combined with the slowing development of new antibiotics over the past few decades [5,6] threatens to further reduce treatment options for previously curable infections, so Western scientific and medical communities are now reengaging to develop phage therapies to help contend with this burgeoning problem.

Phages infect their specific bacterial hosts and in the lytic (or virulent) lifestyle highjack the machinery of the host cell to replicate and ultimately destroy the host, thus simultaneously producing progeny and killing the host. Phages are the most abundant biological entities on earth, with greater than 10^30^ individual virions estimated to be on the planet [7], and a well-described diversity that is still being discovered [8,9]. This vast abundance and diversity of phages in nature provides a ready resource to mine for the selection of phages for a variety of purposes, including not only anti-bacterial therapy but also decontamination, infection control, detection, diagnosis, etc. Phage typing has been used in clinical laboratories to identify species and subtypes of bacteria (e.g., *Salmonella, Bacillus anthracis*, *Staphylococcus*, *Brucella* species) [10].

There are a number of potential advantages of the use of phages as antibacterial therapeutics as have been discussed in numerous reviews and perspectives [11,12,13,14,15], including: (a) since lytic phages have entirely different mechanisms of killing bacteria than antibiotics, they are effective against MDR pathogens and, thus, can also often be used in combination with antibiotics, with frequent synergies; (b) since a phage often can infect only a single bacterial species or a subgroup within a bacterial species, phage therapeutics have high specificity (are narrow spectrum antimicrobials) and, thus, are expected to have little to no effect on normal microflora (in contrast with broad spectrum antibiotics); (c) since phages replicate on the target bacterium, they accumulate precisely where needed to eliminate pathogen cells at the site of infection (are self-replicating drugs); (d) because phages are ubiquitous in the environment and generally easy to isolate, selecting a new phage active against a resistant bacterial strain is straightforward, making new antimicrobial discovery efficient and inexpensive; (e) since phages can encode enzymes that degrade biofilms that can be associated with difficult infections, they can provide access for other antimicrobials to surmount this barrier; (f) phage therapies have been used safely in thousands of humans in the former Soviet Union and Eastern Europe with very few side effects reported; (g) phages with different specificities can be combined in cocktails to address the diversity of a pathogen or to target multiple bacterial pathogens and to address the emergence of resistance; (h) since phages coevolve with their hosts, they can be adapted to newly emergent resistant strains of the host bacterium; (i) since phages replicate to high titer on a bacterial host that is itself self-replicating, they are potentially less expensive to produce than other antibacterials; (j) phages have the potential to stably persist in vivo [16]; and (k) phages generally seem to be weak immunogens and, thus, adverse immunologic responses are unlikely (e.g., enhanced inflammatory response or phage inefficacy due to the potent neutralization by antibodies).

Potential limitations or challenges of phage therapy are: (a) high specificity, since the narrow lytic spectrum/host range of many phages means a single phage cannot be used to treat the diversity within a bacterial pathogen; (b) for some bacterial pathogens, different phage mixtures may be needed to treat the same bacterial disease in different geographical regions; (c) bacterial strains may gain resistance to phages because of alterations in their phage receptor(s) or by other mechanisms and, thus, diminish the effect of treatment (including during the course of a treatment); (d) phage cocktailing requires phage collections that are updated (through discovery or adaptation/engineering) to address newly emerged resistant variants; (e) collecting, maintaining, and using large banks of diverse phages can make safety testing and the regulatory path to an approved therapeutic more difficult and expensive; (f) multiplying the components of cocktails to expand their therapeutic spectrum will make large-scale production of phage therapeutics more complex and expensive; (g) the human immune response to phages used for therapy is not completely understood and could potentially impede efficacy or cause unwanted immune responses [11,12,13,14,15].

## 2. A Brief History of Bacteriophage Therapy

Early observations of bactericidal activity that was possibly attributable to bacteriophages were recorded by Ernest Hankin in 1896 (in the waters of the Ganges and Jumna rivers in India) [17] and Nikolay Gamaleya in 1898 [18], but the clarity of these early observations is open to interpretation [19]. Frederick Twort was the first to hypothesize that a virus was mediating this observed anti-bacterial activity [20]. However, Twort was unable to prove his hypothesis at least in part because of lack of funding, and later the viruses of bacteria were definitively discovered and named bacteriophages by Felix d’Herelle while at the Institut Pasteur in Paris [21]. In 1917, d’Herelle initiated the testing of phage treatments for human patients with severe dysentery at the Hospital des Enfants-Malades in Paris after demonstrating initial safety by ingesting the phage therapeutic himself. Initial efficacy was reported: four patients began to recover within 24 h of treatment. In 1921, Bruynoghe and Maisin reported the first clinical trial in France, in which phages were injected into and around skin lesions from staphylococcal infections and the regression of infection was observed in 24–48 h [22]. In the 1920s, phage therapy expanded in the treatment of thousands of subjects with phage preparations for a variety of infections, including for cholera and bubonic plague in India, and with hundreds of studies on phage therapy in Eastern Europe and the Soviet Union. By the 1930s and early 1940s phage therapeutics were being commercialized, with d’Herelle beginning the production of five phage therapeutics at L’Oréal in Paris. During this period, Eli Lilly Company also began the production of seven phage therapeutic preparations in the US, but this effort was plagued with technical problems and was ultimately abandoned as antibiotic drugs came into production and use. The development and use of phage therapy continued in Poland, Russia, Georgia, and throughout the former Soviet Union [23], while in the West the efficacy of phage therapy remained controversial, since there were very few published studies conducted according to modern clinical research standards, and these were unavailable in English language journals.

An important development that allowed for the re-consideration of phage therapy in the West was the publication of controlled animal studies in the English-language scientific literature in the 1980s [24,25,26] that introduced a new generation of scientists to its potential. For some time now, phage products have been approved for use in the United States to control bacterial contamination in food processing (meat, cheese) [27]. In recent years some Western European countries have begun approved therapeutic use (Belgium, France), while in the United States a number of entities are developing phage therapeutics to bring them into clinical use but these are yet to be approved by the US Food and Drug Administration. Meanwhile, recent FDA-approved “expanded access” experimental phage treatments in cases in the United States [28,29,30,31,32,33] have alerted the news media and public to the safety and potential of phage therapeutics, sparked excitement about phage therapy, and begun to shift the landscape so that more physicians are open to the use of phage treatments in combination with standard of care antibiotics.

## 3. Strategies for Phage Therapeutics

### 3.1. Approaches for Phage Therapeutic Design

The general approaches for phage therapy that have been employed include (a) fixed mixes (or cocktails) of multiple phage components (prét-à-porter, off-the-shelf), (b) cocktails of phages that are periodically modified to add activity against additional circulating clones of the target bacterium or bacteria (modifiable), (c) personalized phage therapy (phage bank, precision, sur-mesure), and (d) in vitro adapted and genetically engineered phage therapeutics [28,34,35,36].

The fixed phage cocktail approach consists of formulating a set combination of phages to address the diversity within a single bacterial pathogen or multiple bacterial pathogens. This approach is employed to develop a phage preparation as an off-the-shelf antibacterial that can be used for treatment and/or prophylaxis [34,35,36,37]. This approach is currently being pursued commercially in Western Europe and the United States because it is most compatible with existing regulatory paths for the clinical development of antimicrobials. Breadth of lytic spectrum (host range) and durability in confronting the challenge of emerging host resistance are key concerns in this approach.

The modifiable cocktail approach begins with the fixed cocktail approach but also includes the periodic addition of or replacement of phage components to broaden the host range of the mix and to address problematic emerging clones of pathogens. This approach allows for the development of off-the-shelf phage therapeutics that can be updated over time to address changing bacterial resistance and epidemiologic realities. Phage therapeutic efforts over decades in the nation of Georgia (also during the time it was a Soviet Republic) have used the modifiable cocktail approach to develop complex phage products, often designed to treat multiple pathogens, and to update them over time [23]. Phage products such as Pyophage and Intestiphage are examples of modifiable cocktails that are still in use today in Georgia and Russia. Metagenomic analysis of these products indicates they are complex mixtures of phages [38,39,40]. This complexity combined with the periodic changes in composition means these products are not directly suited to a traditional regulatory drug development pathway such as the US FDA has established. New regulatory paths are being explored and developed to allow for the modification and/or replacement of phage components in fixed cocktail products that are already approved for human use, since the clones of a bacterial pathogen that are clinically relevant will change over time, and with the long coevolution of phages and their hosts phage resistance patterns will also change over time [41,42].

For the personalized approach, either single phages are selected, or cocktails are compounded to specifically target an infecting strain after its isolation, producing a therapeutic tailored to an individual patient’s infection. A personalized approach to phage therapy has been employed for decades at phage therapy centers such as those at the Eliava Institute in Tbilisi, Georgia, and in Wroclaw, Poland, and this approach is now being pursued in the US by Adaptive Phage Therapeutics, Inc. With this approach speed in culturing the pathogen responsible for the infection and properly selecting a phage or phages with lytic activity against it is a major consideration since the therapeutic is developed in direct response to pathogen isolation and screening. Host range is not a major concern with the personalized approach since the therapeutic is narrowly targeted to address a specific infection strain [28,34,35,43,44]. 

The laboratory adaptation and genetic engineering of phages for therapeutic purposes is likely to be a significant force that will transform phage therapy [13,36,45,46], and the latter is already a commercial focus in the United States for companies such as Locus Biosciences and Armata Pharmaceuticals. In vitro evolution (or phage adaptation, phage training) to expand or optimize host range is an approach to modify and improve phages that has been used extensively in former Soviet countries and Eastern Europe, and was recently described in the Western literature [37,47,48,49]. Alteration of phage host range by this approach can also potentially be used to discover novel molecular mechanisms that can be exploited for genetic modification strategies [50]. Genetic engineering to improve phages as therapeutic candidates has included the alteration or expansion of host range [51], adding factors to enhance therapeutic activity including genes that encode enzymes that degrade biofilms [52] or expressing heterologous genes to increase phage killing potency. In one case, a phage was engineered to overexpress a host gene that suppresses the SOS stress response network in *Escherichia coli* to enhance antimicrobial killing and rescue mice from lethal infection [53]. Engineering has also been used to remove genes to improve the therapeutic potential of phages. In 2019, genetically engineered phages were first used for human therapy, in an expanded access case in which a three-phage cocktail was used to treat a multidrug resistant *Mycobacterium abscessus* infection in a 15 year old cystic fibrosis patient [31]. Phages *ZoeJ* and *BPs* were genetically engineered to remove repressor genes to render them obligately lytic, and thus more suitable for use in humans. Natural phage isolates by themselves are not patentable in the West, while engineered phage products are novel and patentable. Engineering a phage solely to protect intellectual property may not improve its therapeutic potential but is of current commercial consideration.

### 3.2. Designing and Building Fixed Phage Cocktails for Off-The-Shelf Use

Steps that are common to all phage therapeutic development include the collection of therapeutic phage candidates, the characterization of their lytic spectra, further phenotypic characterization, genomic characterization, and the assessment of therapeutic efficacy.

Therapeutic phage collections are assembled based on the approach being pursued, with large libraries of phages needed for the personalized approach, while in contrast a more focused library of selected candidates that meet specific criteria is required for the fixed cocktail approach. Varied sources, such as untreated urban and hospital sewage, environmental waters, or soil samples, can be used for the isolation of new phages by relatively straightforward protocols. The lytic spectra of candidate phages for a fixed cocktail approach will need to be characterized using extensively diverse panels of clinical isolates of the host bacterium in order to determine global applicability, while in the personalized approach lytic spectrum can be focused solely on the patient’s isolate that a therapeutic is being designed for. Phenotypic and genomic characterization is essential to determine whether candidate phages are lytic versus lysogenic, since exclusively lytic phages are selected for therapeutic use. Genomic characterization is also necessary to determine whether a phage is likely to be a transducer, is carrying any bacterial toxin or antimicrobial resistance genes, and to provide a full genomic signature for further use in manufacturing and tracking. Assessing therapeutic efficacy can be as simple as testing the lytic activity against a clinical isolate in the case of personalized therapy, or in the case of developing a fixed therapeutic product cocktail will involve animal modeling to indicate efficacy against a specific medical indication.

Key elements of the design of fixed phage cocktails to provide the necessary host range breadth, efficacy and durability have been described [34,37,54,55,56,57] and include: (a) selection of phages with overlapping host ranges; (b) selection of phages with safe genomic characteristics; (c) selection of phages that infect the same host using different receptors; (d) selection of phages that infect using receptors with a high fitness cost for the host to mutate (resulting in a low resistance rate); (e) selection of phages with anti-biofilm activity; (f) selection of phages that synergize with treatment antibiotics, (g) selection of phages that are non-immunogenic; (h) selection of phages with extended half-life in the therapy recipient (long circulating); (i) selection of phages that are suitable for manufacturing and long-term stability. The chief challenge in designing phage cocktails to have lasting efficacy is in addressing resistance in the bacterial host, since through millions of years of coevolution the host has an array of resistance mechanisms [58], resistance is likely to be encountered during phage therapy [59] but can be overcome using rigorous rational cocktail formulation [56], with combinations of phages optimized to minimize resistance [60].

Optimally designed fixed phage cocktails will possess lytic activity against the majority of strains of a pathogen circulating globally in order to provide the spectrum of activity needed for an off-the-shelf treatment that can be used in combination with antibiotics or alone, or even as a prophylactic. The rational design of broad host range phage cocktails should be focused on developing formulations that will be durable in addressing emerging phage resistance in the pathogen. The design of fixed phage cocktails for different bacterial pathogen species must by nature be centered on the genetic–phenotypic diversity of the pathogen and the characteristics of the lytic phages that infect it. For instance, in the case of *Staphylococcus aureus*, it is widely accepted that relatively few phage components—or even a single phage, in the case of Eliava’s Staphylococcal Phage therapeutic—could be selected to construct therapeutic cocktails that address a significant diversity of the global strains of the pathogen [50,61]. This is because of the relatively broad host range exhibited by Twort and K phages combined with (and perhaps because of) the relatively monomorphic nature of the host species. However, the same is not true for some of the Gram-negative ESKAPE (*Enterococcus faecium*, *Staphylococcus aureus*, *Klebsiella pneumoniae*, *Acinetobacter baumannii*, *Pseudomonas aeruginosa*, and *Enterobacter* spp.) pathogens that are currently being targeted for phage therapy. For example, although phage treatments in experimental *P. aeruginosa*, *K. pneumoniae*, and *A. baumannii* infections in animal models have exhibited efficacy [62,63,64,65], phages specific for *P. aeruginosa*, *K. pneumoniae*, and *A. baumannii* have limited host ranges, usually 5%-50% [65,66,67]. Thus, a mix of several phages in a cocktail must be used to cover the diversity of clinically important strains. Moreover, *P. aeruginosa* [68], *K. pneumoniae* [69], and *A. baumannii* [65] often generate phage-resistant mutants, and the use of phage cocktails with overlapping spectra of lytic activity will help to address this problem. For example, it was shown that the rate of *K. pneumoniae* resistance to each of three phages was quite high (10^−3^, 10^−4^, and 10^−5^ per cell per generation), but the use of a cocktail consisting of the three phages reduced the resistance rate to 10^−6^ [69]. A rationally designed phage cocktail formulation must address significant genetic diversity in the target pathogen. However, a cocktail should also be designed to prevent cross-resistance, i.e., host mutants resistant to one phage in the mix should still remain sensitive to the other components of the cocktail. A strategy in the selection of phages for a fixed cocktail to significantly increase product durability in the face of host resistance will include phage components that use different host receptors for infection [37,41,70], so that if a mutation in the host arises in one receptor to block infection, another phage or phages in the cocktail will still be able to infect. Theoretically, if the phages used in the cocktail described above [69] target three completely different *K. pneumoniae* cell surface receptors, the rate of host resistance to the cocktail could be as low as 10^−12^, making the emergence of mutants resistant to the combined cocktail components statistically improbable. The resistance rates of the host to the individual phages being considered for a cocktail is also a key parameter, since some receptor mutations required to confer phage resistance will have a significant fitness cost for the host bacterium, while others will not. In previous work with *Yersinia pestis*, we identified seven different receptors for the phage components being considered, but also investigated the impact of host mutations on virulence [71]. A comparison of phage-resistant *Y. pestis* mutants in a mouse model of bubonic plague indicated that mutants resistant to certain phages were attenuated, with their lethal dose as much as seven logs greater than the wild-type virulent challenge strain, and providing a significant extension of time to death (greater than 150%) in a highly sensitive model. Such impacts of resistance mutations should certainly be incorporated into phage selection and cocktail design.

### 3.3. Biofilm Degradation and Killing of Bacteria in Biofilms by Phages

Colonization of wounds, surgically implanted materials or catheters by biofilm-producing bacteria such as *P. aeruginosa* and *S. aureus* is a common complication for modern surgical and orthopedic practices. Biofilm formation helps bacteria to evade the patient immune system and enhances antibiotic resistance. A biofilm is a community of microorganisms formed on a biotic or abiotic surface that consist of bacterial cells within an extracellular polymeric substance (EPS)-based milieu composed of polysaccharides, teichoic acids, proteins, and extracellular DNA [72]. Bacterial cells in biofilms are phenotypically different from their planktonic analogs, with reduced motility, distinct gene transcription patterns and a different spectrum of metabolic activity [73,74]. Bacteria within a biofilm EPS form a very tight tissue-like structure that is difficult to remove from a wounds or the surface of a medical device. Bacteria in biofilms are also significantly more resistant to antibiotics than planktonic bacteria, and this complicates the therapy of biofilm-associated infections [75]. The recalcitrance of bacterial biofilms to chemical antibiotic therapy presents a major obstacle to the successful treatment of the biofilm-associated infections that can complicate wound healing and manifest as chronic wounds that lead to amputation or death. Since biofilm formation complicates the treatment of wound infections, several approaches have been developed to control biofilm development. These approaches do not kill the bacteria in the biofilm but rather make the biofilm-bound bacteria planktonic and thus more susceptible to conventional antibiotic therapy. Killing bacteria within a biofilm can be achieved using various antibiotic agents that interfere with essential biological processes or by using different endolysins that disrupt the outer envelope of bacteria. 

A strategy for combating antibiotic resistant biofilm-associated infections such as those by *P. aeruginosa* or *S. aureus* is the use of lytic phages, since it has been demonstrated in a number of laboratories including ours that some phages can efficiently infect and lyse not only planktonic but also biofilm forms of host bacteria. To be effective against biofilms, phages must be able to propagate inside their hosts within biofilm matrix and release their progeny into the surrounding environment for dissemination [76]. This implies that phage candidates effective against biofilms should not only be able to disrupt the EPS matrix of the biofilm but should also provide a robust burst size upon infecting the biofilm bacteria. However, these interactions and the optimal characteristics required for phage anti-biofilm activity still need to be studied and determined. A number of recent publications reported the use of bacteriophages for the control of biofilms caused by *P. aeruginosa* and other bacteria [55,77]. In one publication, marked biofilm clearance was observed, with up to a five-log decrease in bacterial counts [78]. Other authors were less successful, reporting only a 2-log reduction in the number of bacteria in a biofilm formed by *S. aureus* using a specific bacteriophage in combination with sharp debridement in a rabbit wound punch model [79]. Additional approaches included the use of bacteriophages in conjunction with other agents, such as cobalt ion, to reduce biofilm formation or sub-lethal doses of antibiotics in combination with phages to control biofilms formed by *E. coli* in vitro [80]. By expressing polysaccharide depolymerases and lysins, phages penetrate the bacterial capsule or peptidoglycan layer to access host receptors and infect [81]. Since these enzymes could enable penetration and degradation biofilms, phages that encode such enzymes should be included in therapeutic cocktails. Challenges with using phages to degrade biofilms have also been described [82]; this is clearly an area that requires further investigation to enable rational inclusion of phage anti-biofilm capability in therapeutic approaches.

### 3.4. Phage Synergy with Antibiotics

Antibiotics will remain the clinical standard of care treatment of bacterial infections for the foreseeable future despite the challenges of rising antimicrobial resistance and MDR infections, so finding new antimicrobials that work synergistically with them is a key area of near-term focus for drug development. Combinations of phages with antibiotics in treatments can yield synergies that should be exploited to both potentiate the action of antibiotics and also to integrate viable combination therapies into the clinical arsenal, so understanding interactions between phages and antibiotics is an essential aim [83,84,85]. Phage-antibiotic synergy (PAS) has been demonstrated in both Gram-positive and Gram-negative bacteria; sub-lethal concentrations of several classes of antibiotics have a positive effect on the size of phage plaques and efficiency of phage propagation [86,87,88,89]. A mechanism of PAS was recently discovered as lysis delay and significant increase of phage burst size caused by excessive growth and filamentation of bacterial cells and relative shortage of holin supply in response to an antibiotic as a stress-inducing factor [89]. Another more important effect of PAS is synergistic killing of planktonic bacterial cultures [90,91,92,93], and bacterial biofilms in vitro [80,93,94,95,96], as well as enhanced efficacy of combined application of phage and antibiotic in the treatment of colibacillosis in broilers [97], and even therapy of *S. aureus* diabetic foot infections in humans [98]. In a more targeted approach, a lytic phage was selected for *P. aeruginosa* that uses an outer membrane porin as its receptor that is a component of a multidrug efflux system, placing pressure on the host to mutate toward increased drug sensitivity to evade the phage [99]. This represents an approach that aims to re-sensitize MDR pathogens to conventional antibiotics to extend their utility, and selected phages can be paired in combination therapies with the antibiotic(s) to which they increase sensitivity [29]. Notably, some combinations of phages and antibiotics do not provide synergy [87,88,92], thus potential interference must be considered in designing and applying combined antibiotic-phage therapy [100].

## 4. Implications of Human Clinical Trials for Phage Therapeutic Design

Lessons gathered from recent clinical studies of phage therapy [101,102,103,104] will be used to improve trial design but can also inform the design and development of new phage drugs well before clinical trials. Some key learnings for phage product design are listed here. The first efficacious clinical trial reported since the revival of phage therapy in the West used Biocontrol Ltd.’s fixed cocktail of six *P. aeruginosa*-specific phages to treat ear infections [105]. This small-scale trial indicated the potential efficacy of a fixed phage cocktail approach without prior testing of patient isolates to determine susceptibility. A trial of two fixed phage cocktails to treat *E. coli* diarrhea in children in Bangladesh indicated no significant difference between the phage-treated and control groups [106]. However, elements of trial design could have been at play in these results, including the inability to protect the orally delivered phage preparations to transit the stomach [107]. In addition, it was unclear if the 11-phage Nestlé cocktail or the commercial phage cocktail from Russia used in the trial were generally active against circulating *E. coli* strains in southern Asia. For certain pathogens it may be necessary to design phage cocktails with a lytic spectrum for regional strains rather than taking a more global approach. Certainly, if delivering phage orally it would be prudent to design the therapeutic to ensure active phage particles are protected for delivery into the intestine. The Phagoburn trial in France, Belgium, and Switzerland took a multi-pathogen approach to burn infections with a 12-phage fixed cocktail against *P. aeruginosa* and *E. coli* [108]. The *E. coli* arm of the trial had to be abandoned, because the study design for that portion could not be implemented as intended. The *P. aeruginosa* arm of this trial also appeared to have been impaired because of the significant instability of phage potency in the GMP phage cocktail product that decreased by at least four orders of magnitude over the course of the trial. This experience conveys the importance of making sure that phage components are compatible at an early stage in the development of therapeutic cocktail formulations, including testing their stability in mixes. These examples also indicate that simpler cocktail formulation against a single pathogen may be more prudent and effective for early controlled clinical trial forays in the West.

## 5. Conclusions

Bacteriophage therapy is a rapidly growing field with a yet unproven potential for utility in treating antibiotic resistant bacterial infections in the era of burgeoning multidrug resistance and slowing production of new chemical antibiotics. In order to fully ascertain the potential of phage drugs, the collective findings of the scientific community and the long experience of phage therapy in some parts of the world must be leveraged to devise and test new phage therapeutics that can best address the medical needs of our era. Multiple potential approaches to phage therapy are being undertaken that are likely to address different medical indications as they move into clinical practice in the West. Exploiting combinations of phage therapeutics with chemical antibiotics appears to be a promising avenue for near-term clinical development in the West, with stand-alone phage therapy looming as a future potential in an era of decreasing antibiotic use.

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
