# Peer review of "Bacteriophage Therapy: Developments and Directions"

_antibiotics, 2020, doi:10.3390/antibiotics9030135_

Round 1

Reviewer 1 Report

The minireview by Nikolich and Filippov deals with bacteriophage therapy. There are several, recent examples of reviews on the same matter in the literature and it doesn't seem that this one is adding anything to them. Nonetheless, the text is well written and provides information on the matter to non specialists: the editors might like this additional minireview on the topic for their special issue.

If a revised version of the article is to be produced, the following papers should be cited:

- Derek M Lin, Britt Koskella, and Henry C Lin, Phage therapy: An alternative to antibiotics in the age of multi-drug resistance, World J Gastrointest Pharmacol Ther. 2017; 8(3): 162–173. doi: 10.4292/wjgpt.v8.i3.162

- Lucy L. Furfaro, Matthew S. Payne, and Barbara J. Chang, Bacteriophage Therapy: Clinical Trials and Regulatory Hurdles, Front Cell Infect Microbiol. 2018; 8: 376. doi: 10.3389/fcimb.2018.00376

- Gordillo Altamirano FL, Barr JJ. 2019. Phage therapy in the postantibiotic era. Clin Microbiol Rev 32:e00066-18. https://doi.org/10.1128/CMR.00066-18.

- Cui, Z., Guo, X., Feng, T. et al. Exploring the whole standard operating procedure for phage therapy in clinical practice. J Transl Med 17, 373 (2019). https://doi.org/10.1186/s12967-019-2120-z

- Athanasios Kakasis, Gerasimia Panitsa, Bacteriophage therapy as an alternative treatment for human infections. A comprehensive review, International Journal of Antimicrobial Agents, Volume 53, Issue 1, January 2019, Pages 16-21 https://doi.org/10.1016/j.ijantimicag.2018.09.004

On particular subsections:

HISTORY: Nina Chanishvili, Chapter 1 - Phage Therapy—History from Twort and d'Herelle Through Soviet Experience to Current Approaches, Advances in Virus Research, Volume 83, 2012, Pages 3-40

CLINICAL TRIALS (end of the article): Petrovic Fabijan, A., Lin, R.C.Y., Ho, J. et al. Safety of bacteriophage therapy in severe Staphylococcus aureus infection. Nat Microbiol 5, 465–472 (2020). https://doi.org/10.1038/s41564-019-0634-z

The manuscript should also be shortened, avoiding repetition of the same concept. For instance, lines 12-14 of the Introduction section (lines 33-35 of the whole paper) can be removed.

The references should be cited where appropriate rather than being grouped at the beginning of a paragraph. Examples are:

i. History (lines 72-84)

ii. Lines 125-129, refs 33,34,27,35

iii. Lines 153-162. Refs should be spread over the text where appropriate.

References should also be given to support statements as:

i. lines 280-283: "it has been demonstrated". Please, provide a reference.

ii. lines 288-289: "a number of recent publications": just two?

iii. line 290: "some publications...": one?

The quality of presentation would be much improved by the presence of a couple of tables, like one with the commercially available phage cocktails and one with examples of PAS (phage+antimicrobials) and the related references (e.g. lines 310-312. refs. 87-90 Gram+ and Gram- bacteria PAS).

Author Response

Dear Reviewers,

We are very grateful to you for careful reviewing of our manuscript “Bacteriophage Therapy: Developments and Directions” (antibiotics-732045), your useful comments and criticism that helped us to improve the manuscript, which was revised accordingly. All Reviewers’ comments (we numbered them for better traceability) are addressed below point by point in red type.

Best regards,

Mikeljon Nikolich and Andrey Filippov

---------------------------------------------------------------------------------------------------------------------

Reviewers’ comments:

Reviewer #1 (Comments and Suggestions for Authors):

  1. The minireview by Nikolich and Filippov deals with bacteriophage therapy. There are several, recent examples of reviews on the same matter in the literature and it doesn't seem that this one is adding anything to them. Nonetheless, the text is well written and provides information on the matter to non specialists: the editors might like this additional minireview on the topic for their special issue.

Answer: We thank the reviewer for this comment. We think that our minireview summarizes the current knowledge on phage therapy and offers additional insight, especially, into the rational design of fixed therapeutic phage cocktails and the rationale for their use vs. personalized approach in phage therapy.

  1. If a revised version of the article is to be produced, the following papers should be cited:

- Derek M Lin, Britt Koskella, and Henry C Lin, Phage therapy: An alternative to antibiotics in the age of multi-drug resistance, World J Gastrointest Pharmacol Ther. 2017; 8(3): 162–173. doi:

10.4292/wjgpt.v8.i3.162.

- Lucy L. Furfaro, Matthew S. Payne, and Barbara J. Chang, Bacteriophage Therapy: Clinical Trials and Regulatory Hurdles, Front Cell Infect Microbiol. 2018; 8: 376. doi: 10.3389/fcimb.2018.00376.

- Gordillo Altamirano FL, Barr JJ. 2019. Phage therapy in the postantibiotic era. Clin Microbiol Rev 32:e00066-18. https://doi.org/10.1128/CMR.00066-18.

- Cui, Z., Guo, X., Feng, T. et al. Exploring the whole standard operating procedure for phage therapy in clinical practice. J Transl Med 17, 373 (2019). https://doi.org/10.1186/s12967-019-2120-z.

- Athanasios Kakasis, Gerasimia Panitsa, Bacteriophage therapy as an alternative treatment for human infections. A comprehensive review, International Journal of Antimicrobial Agents, Volume 53, Issue 1, January 2019, Pages 16-21 https://doi.org/10.1016/j.ijantimicag. 2018.

09.004

On particular subsections:

HISTORY: Nina Chanishvili, Chapter 1 - Phage Therapy—History from Twort and d'Herelle Through Soviet Experience to Current Approaches, Advances in Virus Research, Volume 83, 2012, Pages 3-40.

CLINICAL TRIALS (end of the article): Petrovic Fabijan, A., Lin, R.C.Y., Ho, J. et al. Safety of bacteriophage therapy in severe Staphylococcus aureus infection. Nat Microbiol 5, 465–472 (2020). https://doi.org/10.1038/s41564-019-0634-z

Answer: There are a hundred review articles on bacteriophage therapy and we could not cite them all in this minireview, but we agree with the reviewer that two of these articles (Furfaro et al., 2018 and Altamirano et al., 2019) are worth citing, so we added these references in the text and in the list of literature (see revised manuscript with tracked changes).

  1. The manuscript should also be shortened, avoiding repetition of the same concept. For instance, lines 12-14 of the Introduction section (lines 33-35 of the whole paper) can be removed.

Answer: We agree and removed this sentence in lines 33-35.

  1. The references should be cited where appropriate rather than being grouped at the beginning of a paragraph. Examples are:
  2. History (lines 72-84)
  3. Lines 125-129, refs 33,34,27,35

iii. line 290: "some publications...": one?

Answer: We concur and made appropriate changes in the text.

  1. The quality of presentation would be much improved by the presence of a couple of tables, like one with the commercially available phage cocktails and one with examples of PAS (phage+antimicrobials) and the related references (e.g. lines 310-312. refs. 87-90 Gram+ and Gram- bacteria PAS).

Answer: We thank the reviewer for this recommendation but would prefer not to expand this minireview with tables that can be found in other review articles.

Reviewer 2 Report

it would be easier to read if the data described were also plotted.

Author Response

Dear Reviewers,

We are very grateful to you for careful reviewing of our manuscript “Bacteriophage Therapy: Developments and Directions” (antibiotics-732045), your useful comments and criticism that helped us to improve the manuscript, which was revised accordingly. All Reviewers’ comments (we numbered them for better traceability) are addressed below point by point in bold type.

Best regards,

Mikeljon Nikolich and Andrey Filippov

------------------------------------------------------------------------------------------

Reviewers’ comments:

Reviewer #1 (Comments and Suggestions for Authors):

Reviewer #2 (Comments and Suggestions for Authors):

  1. Don't feel qualified to judge about the English language and style.

Answer: We thank the reviewer for scoring the quality of the manuscript, especially for giving higher scores for its significance and soundness. Taking into account this option checked by the reviewer, we do not understand exactly why he/she provided a low score for English usage in our manuscript.  That said, we have made some improvements to the grammar and usage in the text.

  1. It would be easier to read if the data described were also plotted.

Answer: We are not certain exactly what the reviewer means by plotting.

Reviewer 3 Report

The manuscript submitted by Nikolich and Filippov summarizes the history and situation of phage therapy. It is well written and shows a very nice overview for those readers that are not familiar with the idea of phage therapy.

I only have some minor suggestions to improve the quality of the manuscript, all of them format related:

  • The authors should check and delete any double spaces present in the manuscript, there are many.
  • Some paragraphs are not justified, please keep the format consistent.
  • Line 22: the "the" of bacteriophages is not required.
  • Line 148: delete "thus" or change the sentence structure.
  • Line 197: delete "professionally" or change it to "exclusively".
  • Please double-check all your references and use the same format, especially for those with a doi number.

Author Response

Dear Reviewers,

We are very grateful to you for careful reviewing of our manuscript “Bacteriophage Therapy: Developments and Directions” (antibiotics-732045), your useful comments and criticism that helped us to improve the manuscript, which was revised accordingly. All Reviewers’ comments (we numbered them for better traceability) are addressed below point by point in bold type.

Best regards,

Mikeljon Nikolich and Andrey Filippov

------------------------------------------------------------------------------------------

Reviewers’ comments:

Reviewer #3 (Comments and Suggestions for Authors):

  1. The manuscript submitted by Nikolich and Filippov summarizes the history and situation of phage therapy. It is well written and shows a very nice overview for those readers that are not familiar with the idea of phage therapy.

Answer: We are profoundly grateful to the reviewer for his/her positive evaluation of our work and for providing comments that helped us to improve the manuscript (see our responses below).

  1. Materials and Methods. The authors should check and delete any double spaces present in the manuscript, there are many.

Answer: We agree and removed all double spaces.

  1. Some paragraphs are not justified, please keep the format consistent.

Answer: We concur and fixed it.

  1. Line 22: the "the" of bacteriophages is not required.

Answer: This is true; we deleted “the” in this line.

  1. Line 148: delete "thus" or change the sentence structure.

Answer: We agree and deleted “thus” and divided this long sentence into two sentences.

  1. Line 197: delete "professionally" or change it to "exclusively".

Answer: We concur and replaced “professionally” with “exclusively”.

  1. Please double-check all your references and use the same format, especially for those with a doi number.

Answer: We fully agree and made all references in the same format, per the instructions to authors of the journal “Antibiotics”, making sure that DOI numbers are included where they are available.

Round 2

Reviewer 1 Report

I do not think this minireview was very much revised.